# Elevated ETV6 Expression in Glioma Promotes an Aggressive In Vitro Phenotype Associated with Shorter Patient Survival

**DOI:** 10.3390/genes13101882

**Published:** 2022-10-17

**Authors:** Zhang Xiong, Shuai Wu, Feng-jiao Li, Chen Luo, Qiu-yan Jin, Ian David Connolly, Melanie Hayden Gephart, Linya You

**Affiliations:** 1Department of Neurosurgery, Huashan Hospital, Shanghai Medical College, Fudan University, Shanghai 200040, China; 2Neurosurgical Institute, Fudan University, Shanghai 200040, China; 3Shanghai Clinical Medical Center of Neurosurgery, Shanghai 200040, China; 4Shanghai Key Laboratory of Brain Function and Restoration and Neural Regeneration, Shanghai 200040, China; 5Department of Human Anatomy & Histoembryology, School of Basic Medical Sciences, Fudan University, Shanghai 200032, China; 6Department of Neurosurgery, Stanford University School of Medicine, Palo Alto, CA 94305, USA; 7Key Laboratory of Medical Imaging Computing and Computer Assisted Intervention of Shanghai, Shanghai 200032, China

**Keywords:** glioblastoma, ETV6, apoptosis, proliferation, migration, invasion, PI3K-AKT, Ras-MAPK

## Abstract

**Background**: GBM astrocytes may adopt fetal astrocyte transcriptomic signatures involved in brain development and migration programs to facilitate diffuse tumor infiltration. Our previous data show that ETS variant 6 (ETV6) is highly expressed in human GBM and fetal astrocytes compared to normal mature astrocytes. We hypothesized that ETV6 played a role in GBM tumor progression. **Methods**: Expression of ETV6 was first examined in two American and three Chinese tissue microarrays. The correlation between ETV6 staining intensity and patient survival was calculated, followed by validation using public databases—TCGA and REMBRANDT. The effect of ETV6 knockdown on glioma cell proliferation (EdU), viability (AnnexinV labeling), clonogenic growth (colony formation), and migration/invasion (transwell assays) in GBM cells was tested. RNA sequencing and Western blot were performed to elucidate the underlying molecular mechanisms. **Results**: ETV6 was highly expressed in GBM and associated with an unfavorable prognosis. ETV6 silencing in glioma cells led to increased apoptosis or decreased proliferation, clonogenicity, migration, and invasion. RNA-Seq-based gene expression and pathway analyses revealed that ETV6 knockdown in U251 cells led to the upregulation of genes involved in extracellular matrix organization, NF-κB signaling, TNF-mediated signaling, and the downregulation of genes in the regulation of cell motility, cell proliferation, PI3K-AKT signaling, and the Ras pathway. The downregulation of the PI3K-AKT and Ras-MAPK pathways were further validated by immunoblotting. **Conclusion**: Our findings suggested that ETV6 was highly expressed in GBM and its high expression correlated with poor survival. ETV6 silencing decreased an aggressive in vitro phenotype probably via the PI3K-AKT and Ras-MAPK pathways. The study encourages further investigation of ETV6 as a potential therapeutic target of GBM.

## 1. Background

Glioblastoma (GBM) is an infiltrating tumor, with tumor cells in close proximity to native brain cell types such as astrocytes, neurons, oligodendrocytes, microglia, and endothelial cells. In collaboration, we previously described the transcriptome of these cell types in healthy and GBM-infiltrated human brain [1]. Astrocytes are considered the dominant cells in GBM. Astrogenesis involves at least four stages during development: radial glial progenitors, proliferating intermediate progenitors, maturing postnatal astrocytes, and mature adult astrocytes [2]. Fetal astrocytes can serve as a scaffold for neuronal migration [3]. We previously found that GBM astrocytes overlapped in their gene expression with fetal astrocytes [1]. Thus, we hypothesized that GBM astrocytes adopted fetal astrocyte transcriptomic signatures involved in brain development and migration programs in order to facilitate diffuse tumor infiltration. To investigate this, we focused on a transcriptional factor called Ets variant 6 (ETV6), the expression of which was high in both GBM tumor core and fetal astrocytes when compared to normal mature astrocytes [1].

ETV6 is a strong transcriptional repressor through its pointed (PNT) domain in biological processes, including the regulation of cell growth and differentiation [4,5,6,7]. It also functions as a transcriptional activator of the ETS-related gene (ERG), which is an important regulator of normal hematopoiesis and which is highly expressed in leukemia [8]. In addition, ETV6 is frequently involved in chromosomal translocations in different cancers. It was first identified as a fusion partner of PDGFRβ in leukemia [9] and subsequently in a variety of fusions with AML1 [10,11], ABL [12], MN1 [13], JAK2 [14,15], EVI1 [16], CDX2 [17], and BTL [18] in hematopoietic malignancies. In addition, it has also been found to be translocated in solid tumors such as congenital fibrosarcoma (ETV6-NTRK3) [19], salivary gland tumors, and secretory breast carcinomas [20,21]. ETV6-NTRK3 transformation increased cyclin D1 expression and relied on two major effector pathways of wild-type NTRK3, the Ras-MAPK and PI3K-AKT pathways [22,23]. Overexpression of ETV6 correlated with poor prognosis for non-small-cell lung cancer [24] and for nasopharyngeal carcinoma [25]. Finally, ETV6 is involved in tumor angiogenesis and is required for vascular development [26,27,28], an important pathological characteristic of GBM [29]. These data suggest that ETV6 is a context-dependent transcription regulator with repressing and/or activating roles, yet its role in glioma has not yet been investigated.

In this study, we investigated the role of ETV6 in GBM tumor progression by first examining its expression in 192 gliomas, which demonstrated high ETV6 expression in GBM and correlation with poor survival. We then examined the effect of ETV6 knockdown on glioma cellular behaviors and found that ETV6 silencing induced apoptosis and reduced proliferation, clonogenicity, migration, and invasion. Finally, we performed RNA sequencing and Western blot to indicate that the role of ETV6 in glioma was likely mediated partially via the PI3K-AKT and Ras-MAPK pathways. Our study encourages further investigation of ETV6 as a potential therapeutic target in GBM.

## 2. Methods

### 2.1. Tissue Microarray and Immunohistochemistry

ETV6 expression was analyzed by immunohistochemical staining on two different cohorts. The first cohort included 2 tissue microarrays from US Biomax (cat#BS17017b and GL805bt). The second cohort consisted of 3 tissue microarrays made in Fudan University affiliated Huashan Hospital. They were reclassified according to the 2021 WHO classification of tumor of the central nervous system, all GBM were confirmed isocitrate dehydrogenase (IDH) wild-type, and 8 of the low-grade gliomas (LGG) were identified as IDH wild-type. Immunohistochemistry was performed using routine methods. The antibodies used were anti-ETV6 antibodies (ThermoFisher Scientific, PA5-35371, 1:150) and goat anti-rabbit biotinylated IgG (Vector Laboratories, PK-4001). Slides were digitally scanned. For semiquantitative analysis of ETV6 staining intensity, staining was arbitrarily scored from 0–3 based on the fraction of ETV6-positive cells (0, <20%; 1, <20–50%; 2, 50–80%; 3, >80%).

### 2.2. Database Interrogations

Publicly available microarray, RNA sequencing, and clinical data of patients with glioma were acquired from the REpository for Molecular BRAin Neoplasia DaTa (REMBRANDT, GSE108476) and The Cancer Genome Atlas (TCGA). The mRNA expression data of ETV6 from REMBRANDT (*n* = 550) were queried via the R2 microarray analysis and visualization platform (https://hgserver1.amc.nl/cgi-bin/r2/main.cgi (accessed on 14 March 2019)). For TCGA (*n* = 1056), and the ETV6 mRNA expression data were directly downloaded from R2 and Xena (https://xenabrowser.net/ (accessed on 14 March 2019)). The molecular pathology of the patients in TCGA was obtained from cBioPortal (https://www.cbioportal.org/ (accessed on 10 March 2019)).

### 2.3. Flow Cytometry

Flow cytometry analyses were performed using a LSR II flow cytometer (BD Biosciences, Franklin Lakes, New Jersey, USA). Cell cycle analysis with BrdU-FITC and 7-AAD was performed according to the manufacturer’s instructions for the BD Pharmingen BrdU flow kit (BD Biosciences). Apoptosis analysis with annexinV-APC and propidium iodide (PI) was performed according to the manufacturer’s protocol for the AnnexinV Apoptosis Detection Kit APC (eBioscience). FlowJo (TreeStar, Ashland, OR, USA) software packages were used for data analyses. Another apoptosis kit used was the AnnexinV-PE/7-AAD Apoptosis Detection Kit APC (Yesen Biotech, Shanghai, China).

### 2.4. Lentiviral Production and Infection

The recombinant lentivirus of small interfering RNA targeting ETV6 (ETV6-siRNA-lentivirus) and control lentivirus (GFP-lentivirus) were commercially available (GeneChem, Shanghai, China). The U251 or A172 cells were infected with lentivirus for 48 h and selected in puromycin of 2 μg/mL. One week later, cells were collected and the expression of ETV6 was determined. The targeting sequence was GCTGCTGACCAAAGAGGACTT.

### 2.5. CCK8 and EdU Assays

The U251 or A172 glioma cells were seeded into 96-well plates at 5 × 10^3^ cells/well and incubated for 12, 24, 36, 48, 72, and 96 h. An amount of 10 μL of CCK8 solution (0.5 mg/mL; Sigma, Saint Louis, MO, USA) was added into each well and incubated at 37 °C for 2 h. Finally, the absorbance was measured at 570 nm to assess cell viability. EdU Apollo 567 Cell Tracking Kit (Rib-oBio, Guangzhou, China) was used for cell proliferation. Hochest 33342 (Sigma, Saint Louis, MO, USA) was used for nuclei staining. The percentage of EdU-positive cells in total cells was calculated based on counting 500 random cells in three independent experiments.

### 2.6. Clonogenic Formation

The U251 or A172 cells were seeded into 6-well plates at 500 cells/well. After 10 days, cells were fixed by cold 4% paraformaldehyde and stained with 1.0% crystal violet. Colony consisting of more than 50 cells was regarded as a single colony and counted manually at ×10 magnification.

### 2.7. Migration and Invasion Assays

For the migration assays, 5 × 10^5^ cells in 1.0 mL of serum-free DMEM were distributed to each transwell insert (BD Biosciences, Franklin Lakes, NJ, USA). After incubation for 24 h, the cells on the upper membrane of the inserts were cleaned, while the cells migrated to the lower membrane of the inserts were fixed in ice-cold methanol at 4 °C and stained with 1.0% crystal violet. To quantify migrated cells, 5 random fields of each membrane were chosen and the averaged counts represented the migration ability. For invasion assays, the transwell-matrigel system was used. The upper inserts were precoated with matrigel (BD Biosciences). The same protocol as the migration assay was then to quantify invaded cells. All calculations were based on three independent experiments.

### 2.8. RNA-Seq Library Preparation and Sequencing

Total RNA was extracted from control or shRNA infected U251 cells by Trizol (Invitrogen, Carlsbad, CA, USA) method. An amount of 1 µg of total RNA was used for RNA sequencing library preparation using TruSeq RNA LT Sample Prep Kit v2 (Illumina, San Diego, CA, USA). Libraries prepared from 3 scramble and 3 shETV6-infected U251 cells were sequenced by 2 × 100 bp using NovaSeq 6000 sequencing system (Illumina, San Diego, CA, USA).

### 2.9. RNA-Seq Read Mapping, Assembly, Differential Expression, and Functional Analysis

The FASTQ files were mapped using HISAT2. The paired end option was selected and the human genome version19 (hg19) was used as the reference genome of the human RNA-seq data. The DESeq2 package was used for differential expression analysis and the Enrichr web tool was used for pathway analysis. The RNA-seq data were deposited in the National Center for Biotechnology Information (NCBI) Gene Expression Omnibus (GSE155052).

### 2.10. Western Blot

Protein lysates were denatured and separated on 10% polyacrylamide gels. After transfer to polyvinylidene fluoride (PVDF) membranes (Roche Applied Science), membranes were blocked in 5% nonfat milk powder for one hour at room temperature and incubated with primary antibodies overnight at 4 °C. After several times of wash with TBST, membranes were incubated with secondary antibody for one hour at room temperature. Finally, the membranes were visualized using the Tanon 5200 Western blotting Detection System (Tanon, Shanghai, China). Primary antibodies used were anti-ETV6 (ThermoFisher Scientific, Waltham, MA, USA; PA5-35371, 1:1000), anti-AKT (Cell Signaling Technology, Danvers, MA, USA; 4691T, 1:1000), anti-p-AKT (Cell Signaling Technology, Danvers, MA, USA; 4060T, 1:1000), anti-p-MEK1/2 (Cell Signaling Technology, Danvers, MA, USA; 9154, 1:1000), and anti-GAPDH (Weiao, Shanghai, China; WB0197, 1:1000). For secondary antibodies, anti-rabbit IgG (Abcam, Cambridge, UK; 7074, 1:10000), anti-mouse IgG (Abcam, Cambridge, UK; 7076, 1:10000), and HRP anti-rabbit (Santa Cruz Biotech., Santa Cruz, CA, USA; 1:10000) were used. Signal was enhanced using an enhanced chemiluminescence technique (Thermo Scientific, Waltham, MA, USA; 34075).

### 2.11. Statistics

Statistical analyses were performed using the two-tailed Student’s *t*-test, and *p* < 0.05 was considered a statistically significant difference. Graphs were generated with Prism 7 (GraphPad Software). Multivariable analysis of survival was performed in the Cox proportional hazards model using SPSS version 25.0.

## 3. Results

### 3.1. ETV6 Was Highly Expressed in GBM and Correlated with Poor Prognosis

According to our previous study, ETV6 was one of the top upregulated genes in both GBM tumor core and fetal astrocytes compared to normal mature astrocytes [1] (Figure 1A). As a transcription factor, its expression was mainly confined to the nucleus (Figure 1B). To investigate its expression in glioma patients, we examined ETV6 expression in 2 tissue microarrays (TMAs) from patients in the United States containing 3 normal brains, 6 GBM-adjacent normal brains, and 89 glioma tumor cores (*n* = 10, 10, 4, 65 for grade I–IV, respectively; US Biomax, BS17017b and GL805bt). Our results showed high ETV6 expression in GBM (Appendix A). To confirm this, we stained another three TMAs from patients in China containing tumor cores from 103 gliomas (*n* = 35, 34, 34 for grade II–IV, respectively) and 5 normal brains. ETV6 expression was significantly correlated with glioma grade (Figure 1C,D). Moreover, we subdivided the 58 gliomas into low (staining intensity of 0–2) and high (staining intensity > 2) ETV6 groups in LGG and GBM, respectively. The expression of ETV6 was inversely correlated with patient survival in IDH-wildtype GBM, but not in IDH-mutant LGG (*p* = 0.48 and 0.02 in LGG and GBM, respectively; Figure 1E–F).

IDH mutations and methylation of the O6-methylguanine-DNA methyltransferase (MGMT) promoter, both important prognostic factors in glioma, occurred in more than 70% of LGG [30] and 50% of grade IV astrocytoma [31], respectively. We used the TCGA and REMBRANDT databases to further validate our findings and determined the association of ETV6 expression with IDH1 mutation or MGMT promoter methylation status. The mRNA expression of ETV6 was positively correlated with glioma grade in REMBRANDT database (Figure 1G). Considering tumor type (oligodendroglioma vs. astrocytoma) also has a significant effect on the survival of glioma patients, and we compared the expression of ETV6 between them using TCGA dataset (Figure 1H). The expression of ETV6 was significantly higher in astrocytoma than oligodendroglioma. High expression of ETV6 was correlated with poor survival in both LGG and GBM from the TCGA database (*p* = 0.025 for LGG, Figure 1I; *p* = 0.034 for GBM, Figure 1J). In contrast, ETV6 expression was associated with MGMT promoter methylation in LGG but not in GBM (Appendix A).

These data showed the ETV6 expression across multiple TMA sources was positively correlated with glioma grade and negatively correlated with survival.

### 3.2. Knockdown of ETV6 in GBM Cell Lines Induced Apoptosis or Attenuated Proliferation

The above results suggest an oncogenic role of ETV6 in GBM. To investigate this, we first performed ETV6 knockdown in U251 glioma cells by small interfering RNA-mediated gene silencing (siETV6) (IDT, TriFECTa^®^ RNAi Kit, Coralville, CA, USA). The knockdown efficiency was confirmed by RT-qPCR at about 70–80% (Figure 2A). Cell growth was significantly decreased (Figure 2B). To test whether this was due to cell proliferation and/or apoptosis deficit, we performed AnnexinV and BrdU staining quantified by flow cytometry. Results showed that ETV6 knockdown led to decreased cell survival, and increased early and late apoptosis (Figure 2C,D) with little effect on other phases of the cell cycle (Figure 2E,F).

We then silenced ETV6 in U251 and A172 glioma cells using lentiviral short hairpin RNA (shETV6) with sufficient knockdown efficiency (Figure 2G,H). One of the key features of cancer cells is survival under nutrient-restricted conditions [32]. Thus, we analyzed cell apoptosis after 7 days of serum deprivation. The results showed that ETV6 silencing led to decreased cell viability and enhanced early apoptosis in U251 (Figure 2I,J) but had little effect in A172 cells (Appendix A). To confirm the effect of ETV6 on the proliferation of glioma cells, cell viability was examined by cell counting kit-8 (CCK8) for 4 days in U251 and A172 cells. ETV6 silencing attenuated cell growth significantly starting from 24 h in both cell lines (Appendix A). To further investigate this, EdU staining was also performed. ETV6 knockdown significantly decreased proliferation in A172 (Figure 2K,L) but not in U251 cells (Figure 2L and Appendix A).

In summary, ETV6 knockdown led to decreased cell viability in GBM cell lines and the effect of increased early apoptosis or attenuated proliferation is cell-line-specific.

### 3.3. Knockdown of ETV6 in GBM Cell Lines Decelerated Clonogenic Growth and Inhibited Migration/Invasion

To further evaluate the effect of ETV6 knockdown on anchorage-independent cell proliferation of GBM cells, we performed a clonogenic formation assay in scramble or shETV6-infected U251 and A172 cells. The number of colonies formed by shETV6-infected cells was markedly reduced (Figure 3A,B), indicating the weakened capability of colony formation upon ETV6 knockdown. In addition, colony size was noted to be much smaller (Figure 3C), possibly due to abnormal cellular morphology (black arrow). We hypothesized that this morphological change may affect migration and invasion capabilities. Therefore, we performed transwell assays with and without matrigel to investigate the effect of ETV6 knockdown on the invasion and migration of GBM cells. Both migration and invasion capacities were decreased significantly upon ETV6 knockdown in both U251 and A172 cells (Figure 3D,E for migration; Figure 3F,G for invasion).

Thus, ETV6 knockdown also led to reduced colony formation, migration, and invasion. Taken together, this in vitro data suggests that ETV6 plays a significant role in the aggressive in vitro phenotype of GBM.

### 3.4. ETV6-Dependent GBM Phenotype may Involve PI3K-AKT and Ras-MAPK Pathways In Vitro

To explore the underlying molecular mechanism of ETV6′s role in GBM progression, we performed RNA-seq analysis on three pairs of scrambles or shETV6-infected U251 cells. A total of 80 upregulated and 64 downregulated genes were differentially expressed upon ETV6 knockdown using the DESeq2 package [33] (Figure 4A, log2FC > 1 or < −1, *p* < 0.05). We then performed pathway analysis using Enrich bioinformatics resources [34,35]. Gene Ontology (GO) enrichment and Kyoto Encyclopedia of Genes and Genomes (KEGG) pathway analysis showed the upregulation of the following pathways—“extracellular matrix organization”, “positive regulation of MAPK cascade”, “NF-kappa B signaling”, and “TNF-mediated signaling pathway” (Figure 4B,C), which was consistent with our prior findings of reduced migration and invasion capabilities (Figure 3). Moreover, “positive regulation of cell motility”, “regulation of cell proliferation”, “PI3K-AKT signaling pathway”, and “Ras signaling pathway” were downregulated (Figure 4B,C), which was also in agreement with our experimental data (Figure 2 and Figure 3). Additionally, the downregulation of PI3K-AKT and Ras signaling upon ETV6 knockdown is consistent with the ETV6-NTRK3 transformation’s effect on Ras-MAPK and PI3K-AKT pathways [22,23]. We then performed Western blot to validate the role of ETV6 in the Ras-MAPK and PI3K-AKT pathways using MEK1/2 and AKT activation, respectively. The results showed that the expressions of p-MEK1/2 and p-AKT were both downregulated in ETV6-silencing U251 cells, which suggested the inhibition of Ras-MAPK and PI3K-AKT pathways (Figure 4D).

Together, the in vitro ETV6-dependent GBM phenotype may be partially explained by the Ras-MAPK and PI3K-AKT pathways.

## 4. Discussion

Our study suggests an oncogenic role of ETV6 in GBM. ETV6 was a poor prognostic factor, with elevated expression that positively correlated with glioma grade and negatively correlated with patient survival. These data were consistent with our in vitro findings, where the knockdown of ETV6 in several glioma cell lines led to increased apoptosis and decreased proliferation, migration, and invasion. This cellular behavior may be partially mediated via the Ras-MAPK and PI3K-AKT pathways.

The ETS family contributes to the tumor microenvironment (TME) by regulating angiogenesis through the tumor-associated endothelium [26]. Several ETS factors participate in endothelial cell physiology, including ETV6 [27,28], which is required for vascular development. ETV6 knockout mice showed embryonic lethality between E10.5 and E11.5, with defective angiogenesis and apoptosis of mesenchymal and neural cells [36]. Additionally, ETV6 is also a selective regulator of adult hematopoietic stem cell survival [37]. As is known, microvascular proliferation is a typical pathological characteristic of GBM [29]. Herein, we believe that ETV6 may play a vital role in the malignant nature of GBM. Studies have revealed chromosomal translocations as the major cause of aberrant ETV6 expression in leukemia [38] and various solid tumors [21,39]. The overexpression of ETV6 has also been reported in non-small-cell lung cancer [24] and nasopharyngeal carcinoma [25], suggesting an oncogenic role of ETV6. However, the low expression of ETV6 is linked to a high risk of colorectal cancer [8]. Mutational inactivation of ETV6 was found in prostate cancer [40]. Therefore, the role of ETV6 in tumor is context-dependent. Tumor suppressors can sometimes act as oncogenes, such as p16 [41]. Besides displaying a well-known transcriptional repressor role, ETV6 also functions as a transcriptional activator of the ETS-related gene (ERG), an important regulator of normal hematopoiesis and highly expressed in ALL and AML [42]. ETV6 also interacts with histone acetyltransferase TIP60, a transcriptional coactivator [43]. Thus, it is very likely that ETV6 is a context-dependent transcription regulator with repressing and/or activating roles.

Herein, we examined ETV6 expression in 192 cases of glioma from Chinese and US cohorts, followed by validation in 1071 cases of glioma from public datasets of TCGA and REMBRANDT. ETV6 was consistently highly expressed in GBM, with its expression positively correlated with glioma grade and negatively correlated with survival. These data strongly suggest that ETV6 is a poor prognostic factor in glioma. Further, the silencing of ETV6 in glioma cell lines induced apoptosis (Figure 2), attenuated proliferation (Figure 2), inhibited clonogenic growth, migration, and invasion (Figure 3). RNA-seq-based analysis elucidated the underlying mechanism of ETV6 silencing in U251 cells. Our results are consistent with the known effect of the ETV6-NTRK3 oncogenic fusion protein on PI3K-AKT and Ras signaling [22,23]. High expression of ETV6 in GBM may regulate the Ras pathway and contribute to the malignant phenotype.

These results further demonstrated the tight relationship between ETV6 and the malignant nature of glioma in vitro. To verify the above hypothesis, CRISPR and other mutagenesis techniques for KO of ETV6 can be performed in vivo to study glioma evolution in autochthonous models [44,45]. Recent work has demonstrated cooperativity between EGFR and multiple components of the Ras/PI3K pathways, including Pten, Nf1, and Spred1, in driving GBM evolution [46]. We then analyzed the correlation of ETV6 and EGFR and VEGF using GEPIA (http://gepia.cancer-pku.cn/index.html (accessed on 3 January 2021)) [47]. Interestingly, the expression of ETV6 is significantly positively related to the expression of EGFR and VEGF in glioma, especially in GBM. We suspected that ETV6 may closely associate with the angiogenesis of GBM.

Inhibitors targeting ETV6 itself have not been well characterized, although several inhibitors, including PKC412 (midostaurin) [48], ALK inhibitor crizotinib [49,50], and TRK inhibitors, larotrectinib and PLX7486 [51] for ETV6-NTRK3 fusion protein, have been reported. Among these, larotrectinib has been used in a patient with refractory pediatric secretory breast cancer with ETV6-NTRK3 fusion [52] and showed a marked and durable antitumor effect in 55 patients with NTRK fusion-positive cancer.

## 5. Conclusions

ETV6 expression is an unfavorable prognostic factor in glioma. We have shown that its inhibition attenuated the GBM malignant phenotype in vitro probably via the Ras-MAPK and PI3K-AKT pathways. Our data suggest that ETV6 may be a putative therapeutic target for glioma that merits further investigation.

## Figures and Tables

**Figure 1 genes-13-01882-f001:**
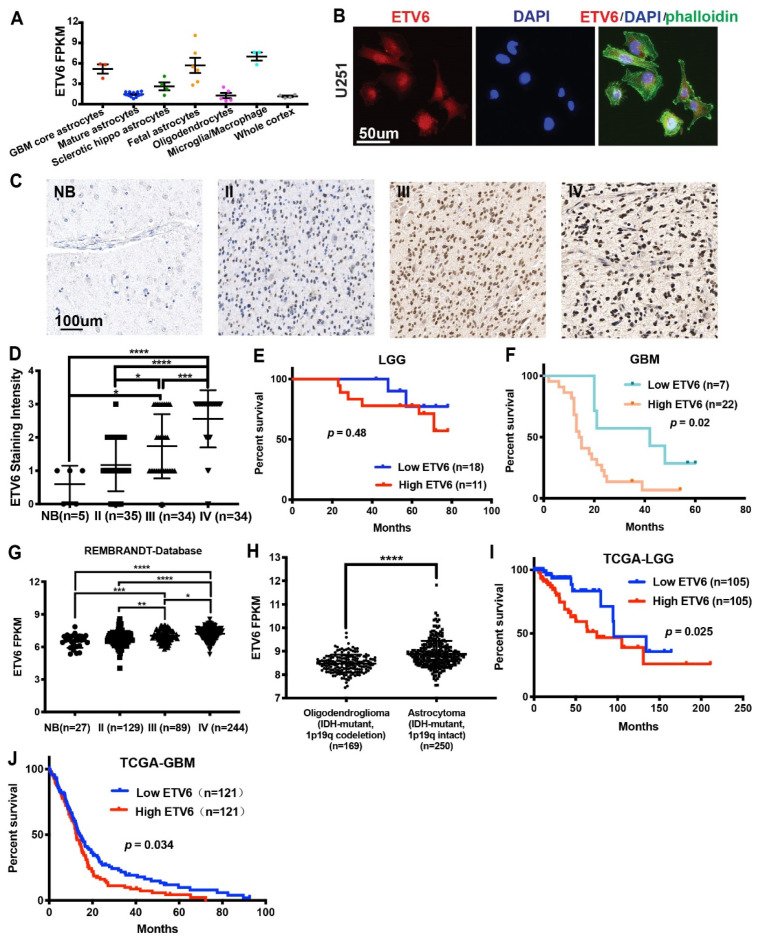
ETV6 expression was positively correlated with glioma grade and negatively correlated with patient prognosis. (**A**) ETV6 was one of the top upregulated genes expressed in both GBM astrocytes and fetal astrocytes compared to normal mature astrocytes as revealed by cell-type-specific RNA-seq [1]. (**B**) Immunofluorescence staining of ETV6 in U251 glioma cells showed its expression mainly in the nucleus. Scale bar, 50 μm. (**C**) Representative images of ETV6 staining of normal brains, grade II, III, and IV gliomas from tissue microarrays were shown. Scale bar, 100 μm. (**D**) The staining intensity of ETV6 was scored as 0–3 on normal brain and grade II–IV glioma (*n* = 5, 35, 34, 34, respectively). ETV6 staining was positively correlated with the grade of glioma. (**E**) ETV6 expression was not significantly correlated with the survival in IDH-mutant LGG (*n* = 18 for low ETV6 vs. *n* = 11 for high ETV6 group, *p* = 0.48). (**F**) ETV6 expression was negatively correlated with patient outcome in IDH-wildtype GBM (*n* = 7 for low ETV6 vs. *n* = 22 for high ETV6 group, *p* = 0.02). (**G**) ETV6 mRNA expression was positively correlated with the WHO grade of glioma in the REMBRANDT database; *n* = 27, 129, 89, 244 for normal brain and grade II–IV glioma, respectively. (**H**) ETV6 mRNA expression was significantly higher in astrocytoma (IDH-mutant and 1p19q intact) than oligodendroglioma (IDH-mutant and 1p19q codeletion) from TCGA; *n* = 250 and *n* = 169 for astrocytoma and oligodendroglioma, respectively. (**I**,**J**) High ETV6 expression was a dismal prognostic factor in IDH-mutant LGG (*n* = 105 for low and high ETV6 groups, respectively, *p* = 0.025) and IDH-wildtype GBM (*n* = 121 for low and high ETV6 groups, respectively, *p* = 0.034) from TCGA (*, *p* < 0.05; **, *p* < 0.01; ***, *p* < 0.001; ****, *p* < 0.0001).

**Figure 2 genes-13-01882-f002:**
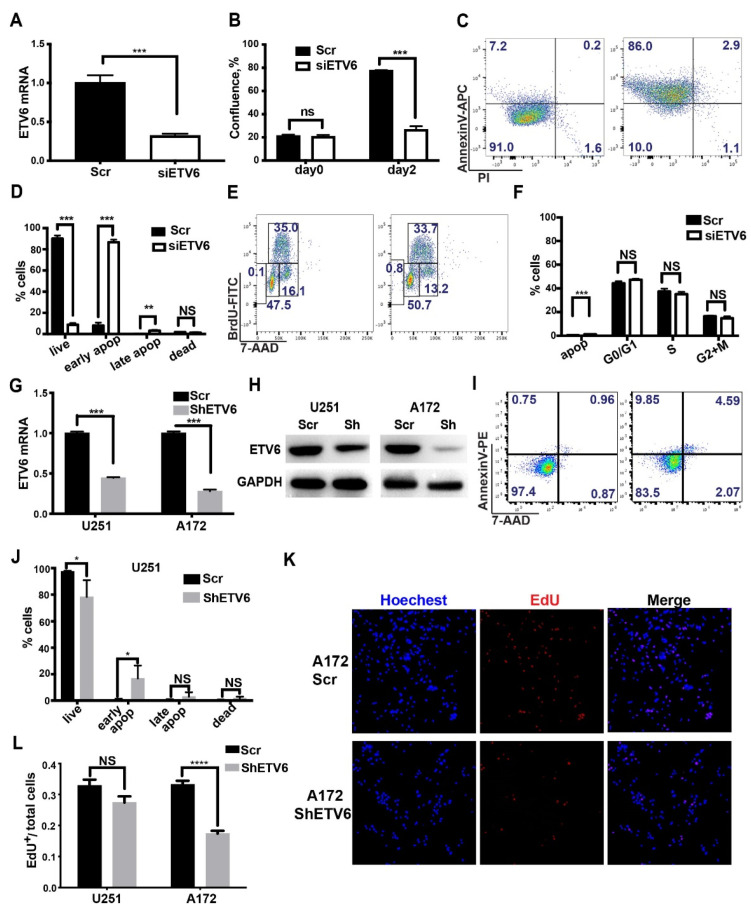
ETV6 knockdown led to increased apoptosis and decreased proliferation in GBM cell lines. (**A**) ETV6 knockdown efficiency reached 70–80% in siETV6 transfected U251 cells, as determined by qRT-PCR. (**B**) Cell confluence was significantly decreased in siETV6 transfected U251 cells as monitored by IncuCyte. (**C**) Control or siETV6 transfected U251 cells were stained with AnnexinV-APC and PI for apoptotic analysis. Representative flow images are shown. (**D**) Quantification showed that ETV6 knockdown led to decreased survival and increased early and late apoptosis. (**E**,**F**) Cell proliferation and cell cycle were minimally affected in siETV6 transfected U251 cells, as revealed by BrdU-FITC and 7-AAD flow cytometric analysis. Representative images (**E**) and quantification of cell ratios in different phases of cell cycle are shown (**F**). (**G**,**H**) Knockdown efficiency of shETV6-infected U251 and A172 glioma cells was determined by qRT-PCR and Western blot, respectively. GAPDH was used as the internal control. (**H**) was cropped from original files named “ETV6 gel” and “GAPDH gel”. (**I**,**J**) Scramble or shETV6-infected U251 cells were stained with AnnexinV-PE and 7-AAD for apoptotic analysis. Representative flow images (**I**) and quantification (**J**) are shown. Consistently, ETV6 knockdown by shRNA led to increased early apoptosis and decreased survival in U251 cells. (**K**,**L**) EdU assay revealed that shETV6-infected A172 cells had decreased DNA synthesis but not shETV6-infected U251 cells. Scale bar, 100μm (*, *p* < 0.05; **, *p* <0.01; ***, *p* < 0.001; ****, *p* < 0.0001; NS, not significant).

**Figure 3 genes-13-01882-f003:**
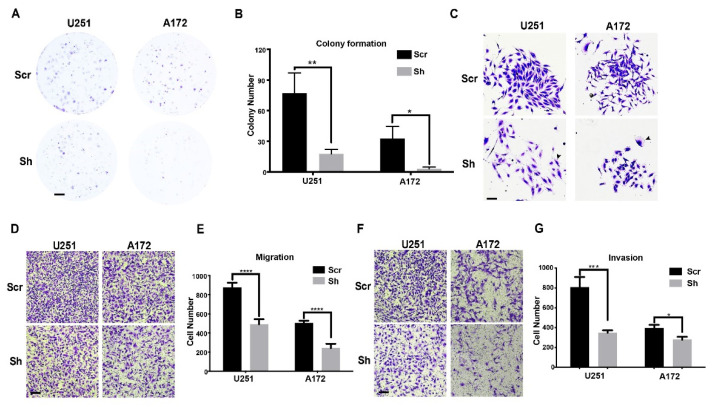
ETV6 knockdown by shRNA decelerated clonogenic growth and inhibited migration and invasion of glioma cells. (**A**,**B**) Colony formation assay was performed using scramble or shETV6-infected U251 and A172 glioma cells. Colony number was significantly reduced in shETV6-infected U251 and A172 cells. Representative images (**A**) and quantification (**B**) are shown. Scale bar, 2 mm. (**C**) enlargement of (**A**) showed that colonies formed by shETV6-infected U251 and A172 cells were smaller and less compact. Scale bar, 200μm. (**D**,**E**) shETV6-infected U251 and A172 cells had decreased migration capability as revealed by transwell assay. Representative images and quantification are shown. Similarly, transwell assay with matrigel revealed that ETV6 knockdown also reduced invasion in U251 and A172 cells (**F**,**G**). Scale bar, 100 μm (*, *p* < 0.05; **, *p* <0.01; ***, *p* < 0.001; ****, *p* < 0.0001).

**Figure 4 genes-13-01882-f004:**
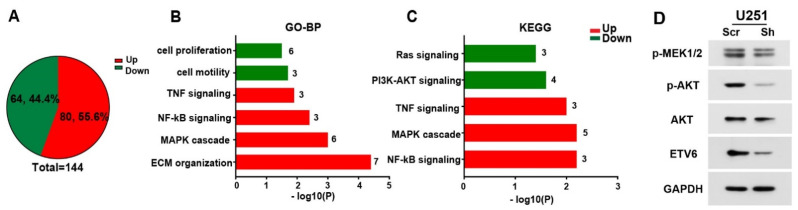
Differentially regulated transcriptional network in ETV6-silenced GBM cells. (**A**) A total of 80 upregulated and 64 downregulated genes were differentially expressed upon ETV6 knockdown via shRNA in U251 cells. p < 0.05, log2FC > 1 or < −1. (**B**,**C**) Gene ontology (GO) enrichment and KEGG pathway analyses were performed using Enrichr. Representative upregulated (red) and downregulated (green) terms are shown. Gene numbers involved in each term are indicated. (**D**) Immunoblotting of validation on Ras pathway (p-MEK1/2) and PI3K-AKT pathway (p-AKT and AKT) in ETV6 silenced U251 cells.

## Data Availability

The RNA-seq data were deposited in the NCBI-GEO repository (GSE155052).

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
