# Peer review of "Elevated ETV6 Expression in Glioma Promotes an Aggressive In Vitro Phenotype Associated with Shorter Patient Survival"

_genes, 2022, doi:10.3390/genes13101882_

Round 1

Reviewer 1 Report

ETV6 is a novel biomarker and the authors showed that ETV6 contributes to aggressiveness in GBM cells. The following issues must be addressed.

Major:

1. According to WHO 2021 classification of brain tumors, IDH mutant grade 4 gliomas are no longer called GBM. They are Astrocytoma grade 4. The authors mention that 50% of the GBMs are IDH mutant. The reference for this was published in 2014 when the new guidelines were not published. This should be updated.

2. The authors do not mention how the TMA case diagnosis was made. Which classification was followed? Was IDH wild type status confirmed before assigning the diagnosis of GBM? Were all LGGs IDH mutant? Were there any grade 4 tumors which were IDH mutant? This must be clarified.

2. If LGG group was a mixture of IDH mutant and wild type gliomas, multivariate survival analysis must be performed in order to conclude that ETV6 is an independent prognostic marker.

3. The TCGA data must be reanalyzed with IDH wild type grade 4 gliomas only considered as GBM and multivariate survival analysis must be performed including IDH mutation as a variant for lower grade gliomas.

4. The tumor type (oligodendroglioma vs astrocytoma) also has a significant affect on survival of the patients. This was not included in any analysis in this paper. 

5. Since the article stresses on the conclusion that ETV6 expression is associated with short patient survival, the histomolecular characterization of the tissues studies and re-analysis of TCGA data to match the updated WHO classification is very essential to draw such a conclusion.

Minor:

Grammatical errors and sentence construction errors are present throughout the manuscript. Please edit the manuscript to correct for errors.

Author Response

please find the attached word file for point-to-point responses. Thanks.

Reviewer 2 Report

 In the current manuscript, entitled "Elevated ETV6 expression in glioma promotes an aggressive in vitro phenotype associated with shorter patient survival”, Zhang et al., investigated the potential role of ETV6 played in GBM tumor progression. They found positive correlation of ETV6 expression level with GBM tumor in two microarray data sets. By analyzing public available data from TCGA and REMBRANDT, they further validated their conclusions. ETV6 knock down in glioma cells led to increased apoptosis and decreased proliferation, clonogenicity, migration, and invasion. Finally, they revealed down-regulation of RAS and PI3K-AKT pathway in ETV6 knockdown group by RNAseq and Western blot.

Overall, the authors provided extensive and convincing data to support the conclusion. In their 2016 Neuron paper, they found ETV6 was highly expressed in GBM astrocyte and fetal astrocyte, but not in mature astrocyte and oligodendrocyte. This indicates ETV6 may have proliferation promoting function in GBM. In the current study, they provided detailed bioinformatics and functional data to show ETV6 may serve as oncogene in GBM.

There are two minor questions:

1. Why lentiviral-shETV6 shows different effect on U251 and A172 in terms of apoptosis and EDU incorporation in Fig2? Does this relate to ETV6 level in different experiment conditions? Especially, shETV6 decreases proliferation, clonogenicity, migration, and invasion in both cell lines in Fig3.

2. Could the authors double check their WB image of p-Akt in Fig4D? Is it flipped horizontally? Based on their RNAseq data and reference 23, ETV6 knockdown should decrease p-Akt level.
